# Graphene Oxide: Opportunities and Challenges in Biomedicine

**DOI:** 10.3390/nano11051083

**Published:** 2021-04-22

**Authors:** Pariya Zare, Mina Aleemardani, Amelia Seifalian, Zohreh Bagher, Alexander M. Seifalian

**Affiliations:** 1Department of Chemical Engineering, University of Tehran, Tehran 1417466191, Iran; paria.zare95@gmail.com; 2Biomaterials and Tissue Engineering Group, Department of Materials Science and Engineering, Kroto Research Institute, The University of Sheffield, Sheffield S3 7HQ, UK; maleemardani1@sheffield.ac.uk; 3Watford General Hospital, Watford WD18 0HB, UK; a.seifalian@nhs.net; 4UCL Medical School, University College London, London WC1E 6BT, UK; 5ENT and Head and Neck Research Centre and Department, Hazrat Rasoul Akram Hospital, The Five Senses Health Institute, Iran University of Medical Sciences, Tehran 1445413131, Iran; 6Nanotechnology and Regenerative Medicine Commercialisation Centre (NanoRegMed Ltd.), London BioScience Innovation Centre, London NW1 0NH, UK

**Keywords:** graphene, graphene oxide, functionalization, interface, stem cells, cell adhesion, carbon, human organs, 3D scaffold

## Abstract

Desirable carbon allotropes such as graphene oxide (GO) have entered the field with several biomedical applications, owing to their exceptional physicochemical and biological features, including extreme strength, found to be 200 times stronger than steel; remarkable light weight; large surface-to-volume ratio; chemical stability; unparalleled thermal and electrical conductivity; and enhanced cell adhesion, proliferation, and differentiation properties. The presence of functional groups on graphene oxide (GO) enhances further interactions with other molecules. Therefore, recent studies have focused on GO-based materials (GOBMs) rather than graphene. The aim of this research was to highlight the physicochemical and biological properties of GOBMs, especially their significance to biomedical applications. The latest studies of GOBMs in biomedical applications are critically reviewed, and in vitro and preclinical studies are assessed. Furthermore, the challenges likely to be faced and prospective future potential are addressed. GOBMs, a high potential emerging material, will dominate the materials of choice in the repair and development of human organs and medical devices. There is already great interest among academics as well as in pharmaceutical and biomedical industries.

## 1. Introduction

Trauma, aging, and disease cause damage to tissue, which can result in loss of function. In the early stages, self-healing can be effective, but with substantial injury, the organ cannot restore itself sufficiently; thus, organ repair or replacement (autograft or allograft) is recommended, albeit with some challenges. The most critical challenges facing organ replacements are organ shortages, grafts, transplant rejection, high cost, inflammation, infection, and also death in some cases [1]. Alternatively, tissue engineering has attracted considerable attention due to its significant potential for tissue restoration. Tissue engineering is a combination of active molecules, cells, and also a scaffold that accurately mimics the extracellular matrix [2]. An optimal scaffold should be nontoxic, porous, provide mechanical strength and proper cell attachment, and, consequently, induce cell proliferation and differentiation [3].

Carbon-based nanomaterials have recently boomed in biomaterial applications. Graphene is an important new addition to these carbon family materials due to its unique properties. Researchers in tissue engineering have already extensively investigated graphene-containing structures, specifically for bone, neuronal, cardiac, skin, cartilage, and dental tissue regeneration, as highlighted in graphical abstract. Carbon-based nanomaterials have recently boomed in biomaterial applications. Graphene is an important new addition to these carbon family materials due to its unique properties. Researchers in tissue engineering have already extensively investigated graphene-containing structures, specifically for bone, neuronal, cardiac, skin, cartilage, and dental tissue regeneration, as highlighted in Figure 1.

In this review, graphene oxide (GO), reduced graphene oxide (rGO), and functionalized GO (FGO) are noted among the various allotropic forms of carbons due to their significant surface area, strength, light weight, chemical stability, enhanced cell adhesion, proliferation, differentiation, and application in the repair of tissue [4,5]. The basic structure of all carbon allotropes is graphene; it is a single sheet of sp^2^-hybridized carbon atoms arranged in a hexagonal lattice-like honeycomb. Valuable research has completely described the synthesis method, but in brief, the synthesis approach was divided into top-down and bottom-up strategies. In the top-down approach, graphite is treated by exfoliation (mechanical or electrochemical). The bottom-up approach is a developed approach, which contains epitaxial growth, chemical vapor deposition, and the dry ice method for graphene material production [6,7,8] (Figure 2).

Two-dimensional (2D) graphene lattice structures have some shortcomings, such as an unstable chemical structure and limited active sites for interacting with other molecules or nanomaterials; hence, it has some incompatibility. The chemical modification of graphene and the production of GO resolves some of these problems. GO maintains graphene’s atomic configuration and only has carboxyl groups (-OOH) on the edge of its structure, as well as epoxy (-O) and hydroxyl (-OH) groups on the basal plane. By reducing the amount of oxygen in GO through thermal, chemical, or UV exposure processes, rGO is produced. Most often, the reduction of GO does not complete, and some oxygen groups remain because not all sp^3^ bonds could return to the sp^2^ structure [10]. rGO promotes the cell differentiation and mechanical properties of scaffolds, although there is no apparent effect on scaffold hydrophilicity [11]. Thus, the repair and replacement of organs using tissue engineering strategies have focused on GO, FGO, and rGO more than graphene. The aim of this research was to highlight the physicochemical and biological properties of GO-based materials (GOBMs) and critically review the literature over the last three years on the applications of these materials in the repair and development of human organs.

## 2. Graphene and Its Physicochemical Properties

The distinctive properties of graphene are derived from its particular crystal lattice structure. Within this, the bonding between each carbon atom is hybridized sp^2^ with the addition of π orbitals. In each unit cell of graphene, two π orbitals exist that are dispersed to form two π bonds, both of which could be known as bonding and antibonding [12]. This arranged lattice is a fundamental building block for all graphitic materials in various dimensions, namely (1) zero-dimensional (0D), e.g., carbon dot, fullerenes and nanodiamonds; (2) rolled one-dimensional (1D), e.g., carbon nanotubes; (3) two-dimensional (2D), e.g., graphene and GO; and (4) stacked three-dimensional (3D), e.g., graphite [13]. The graphene family structure also results in an exceptional surface-to-volume ratio, high intrinsic mobility, unparalleled thermal conductivity, and excellent electrical, optoelectronic, and mechanical properties that have paved the way due to being attractive technological tools [12,14]. Graphene is renowned as one of the most robust materials known to humans, and it is found to be 200 times stronger than steel [15]. In GO, hydrogen bonding forms between hydroxyl and epoxy groups and weak interactions with other groups. The existence of the carboxylic acid group offers a negative surface charge (hydrophilic section); therefore, GO has stability in different polar solutions (particularly water), while graphene is inclined to aggregation. Moreover, owing to free surface π electrons from unmodified graphene (hydrophobic section), GO has an amphiphilic structure that could act as a surfactant. Graphene is hydrophobic, and GO, in comparison with graphene, could be hydrophilic or hydrophobic depending on the chemical and functionalization of the surface chemistry. These characteristics make GO the most important derivative of graphene, which possesses an easy process and a high affinity to accommodate biomolecules. The enhancement of chemical reactivity and graphene stability in solution is intertwined with the presence of reactive oxygen functional groups. Disrupted sp^2^ reduces its mechanical, electrical, and thermal properties [12,14]. Although rGO has less oxygen content, hydrophilic functional groups, or surface charge, through the modification of noncovalent interactions (e.g., van der Waals interactions and π–π stacking), the physical adsorption of both polymers and small biomolecules onto its basal plane is enhanced remarkably [16].

## 3. Graphene Oxide and Its Biological Properties

It has been proven that most GO and derivatives are cytocompatible in vitro and in vivo. However, the physicochemical properties of 2D materials, such as structure, shape, size, surface functionality, concentration, and aggregation state have an essential impact on cellular behavior. Graphene, with its sharp edge properties, has the potential to cause cell damage during the penetration of cell membranes. Its aggregation can also lead to cytotoxicity. Graphene at the nanoscale, when <100 nm, results in cytotoxicity, inflammation, and even genotoxicity (due to facing less steric hindrance). In contrast, graphene with functionalized groups (i.e., GO, FGO such as the amine group, and rGO) is easily internalized by cells (especially in nano sizes), in addition to causing more irregular cell membrane perturbation [12,17]. GO and its derivatives have been ascertained to have specific antibacterial properties, which are also emphasized in tissue engineering applications [18]. The antibacterial activity of GO is related to various mechanisms, including membrane stress, oxidative stress, entrapment, the basal plane, and the photothermal effect. GO has sharp edges that damage the cell membrane, meaning it could, in turn, lead to bacterial cell mortality via the membrane stress mechanism. The structure of GO allows it to act as an electron acceptor; thus, in the vicinity of bacteria, the abstraction of electrons within the membrane occurs, compromising membrane integrity and killing bacterial cells (particularly *P. aeruginosa* and *S. aureus*). GO and rGO, owing to the existence of functionalized groups, can alter the partial pressure of intracellular oxygen, which results in oxidative damage that destroys the bacterial cell internal composition, particularly *E. coli*, through the deactivation of their proteins and lipids, which eventually leads to cell death. GO has illustrated synergistic effects with laser energy; hence, it has been used for photothermal therapy, directly enhancing its antibacterial activity [19]. Another fascinating property that GOBMs possess is antioxidant activity, and sp^2^ carbons play an essential role in scavenging radicals by radical adduct formation and electron transfer. Because of this characteristic, these biomaterials can effectively scavenge radicals and protect cells from high levels of oxidative stress [20]. Graphene, being nonbiodegradable (except FGO), presents serious concerns for potential toxicity, immune response, and environmental hazards [21]. It is reported that GO is susceptible to biodegradation by oxidative attack through hydrogen peroxide and horseradish peroxidase. Therefore, many attempts, such as the fabrication of nanocomposites, have been carried out to accelerate GO biodegradation, as the degradation rate of biomaterials (i.e., scaffold) must be compatible with the rate of tissues and organs [15,22,23].

## 4. Development of Tissues and Organs Using Graphene-Based Materials

GO and its derivatives have been added to other biomaterials either as filler to enhance their properties or as a conjugate during synthesis. The latter makes the nanocomposite materials much stronger. Below are examples of the applications of graphene derivatives in the repair and replacement of human organs (Table 1).

Having drastic proliferation and differentiation of stem cells into the specific tissue lineage, considered as the most critical features of an ideal scaffold. As mentioned before, GBMs have considerable potential for cells stimulating in stem cell-based therapy. In Table 2, we summarized the most common stem cells utilized for each tissue.

### 4.1. Nerve Muscle and Cardiac Tissue Engineering

Electrical signals and external stimuli enhance tissue regeneration for excitable tissue, specifically nerve, muscle, and cardiac tissue. Depolarization and repolarization take place between the two sides of both nerve and muscle cell membranes under an action potential, which leads to their contracting activity and response to electrical signals. Conductive materials, such as GOBMs, with similar electrical conductivity to native tissue, are considered to be promising scaffolds that stimulate cell proliferation and differentiation in stem-cell-based therapy, with decreased cytotoxicity and improved mechanical properties. They are better electron transmitters in comparison to other electronic materials, such as carbon nanotubes [30]. Much effort has been made to repair neuronal injuries, assisted by GOBMs to provide the signaling pathways between cells [36]. Despite the acceptable cellular outcome of GO film in neural tissue regeneration, the precoating of GO by polymers (e.g., collagen, laminin, poly-L-lysine (PLL), or PDL) could improve cell adhesion. There is also no preference between utilizing graphene or GO film because neural differentiation is affected by different fabrication approaches and different cell types [37]. Furthermore, Zhang et al. demonstrated the enhanced neural differentiation of a GO mat compared to an rGO mat by immersing the modified glass in 1.5 mg/mL of GO aqueous solution and then reduced the oxygen-containing groups to obtain an rGO mat [38]. The outstanding properties of the 3D structures of GO, such as electrospun mats, foam, hydrogels, and layer-by-layer casting (LBLC), in addition to their supportive role in cell viability and neural cell differentiation, are its sufficient porosity, which facilitates nutrient exchange and a high surface/volume ratio, which provides highly conductive pathways for charge transport in neural networks [37,38]. Biodegradable materials, including polycaprolactone (PCL) and poly(lactic-co-glycolic acid) (PLGA), are the most popular polymers along with GOBMs proposed in electrospinning nerve tissue engineering in recent years [25]. In the electrospinning design, achieving an optimal GO concentration is substantial, as GO concentration affects the physicochemical structure, mechanical properties, and eventually the biological properties of the material. Thus, manufacturing a scaffold with a varied GO concentration is necessary. GO concentration can affect the fiber diameter in electrospun mats. Subsequently, GO concentration can affect cellular colonization in the scaffold, which is another important parameter in nerve tissue engineering [39]. Apart from the important effect of GO concentration, the dryness or wetness state of the scaffold also affects its final properties. A new study in 2021 revealed that the conductivity, metabolic activity, and cell proliferation of the scaffold, which is a combination of silk fibroin and GO (or rGO), increases after hydration [40]. Due to the limited dispersibility of graphene-based materials (particularly pure graphene) in solvents, the conventional electrospinning method is not considered to be an efficient strategy. Thus, in novel approaches, GOBMs are coated on the surface of the nanofibrous scaffold instead of encapsulated in fibers, with no damaging effect on the nanofibrous structure [25,37,41]. Juan and coworkers demonstrated that Schwann cells can upregulate various myelin gene expressions and also release many neutrophils to support nerve regeneration in rGO-coated electrospun mats. The nanofibrous mat was saturated at the amount of 1.4% rGO, and the electrical conductivity was revealed to be 4.05 × 10^−2^ S/m [25]. The LBLC method was utilized to fabricate the 3D porous graphene conduit, considered an altered electrospinning method. In this method, adhesive macromolecules coated single and multilayered GO/PCL nerve conduits (Figure 3A). Via an integrated 3D-printing and LBLC procedure, the nanoscaffolds revealed a perfect cellular behavior. Due to the potency of GOBMs in nerve regeneration, there are some endeavors to develop a new class of these biomaterials, particularly in inorganic conditions. A unique hybrid scaffold made of a transition metal, synthetic Mo_0.25_Co_1.257_W_0.25_S_3_ hybridized with GO (MCWS/GO), was prepared, which led to successful nerve regeneration and recovery within 2–5 weeks after MCWS/GO transplantation [42].

Similar to the nerves, muscles (skeletal, cardiac, and smooth) have an electrical stimuli-responsive characteristic; thus, the utilization of conductive biomaterials was proposed. There was no difference in cell adhesion and cellular growth in a random and aligned GO electrospun mat, which was modified with oxygen plasma [43]. GO-polyurethane (PU) foam is considered to be a beneficial scaffold for myogenesis in skeletal tissue engineering. This scaffold is fabricated by the dip-coating method with a biocompatible concentration of 10 μg/mL for GO. The results showed enhanced spontaneous myogenic differentiation without any myogenic factors [44]. Xin et al. designed a muscle-inspired, self-healing, conductive, and also self-adhesive hydrogel through the combination of chitosan (CS), GO (lower than 0.75 mg/mL), and polydopamine (PDA). PDA has a great binding ability to a variety of materials. Chemically crosslinked networks enhanced electrical conductivity and also increased adhesive features caused by the presence of DA in the hydrogel (Figure 3C). The electrical conductivity of the hydrogel was revealed as 1.22 ms/cm, matching the native myocardium. The cell results also showed increased stem-cell-derived fibroblast and cardiomyocyte (CM) adhesion [29].

As mentioned, cardiac muscles are electrically conductive (0.03 to 0.6 S/m^3^); hence, conductive biomaterials (especially with polymeric substrates) are potential candidates that can mimic the native extracellular matrix (ECM) for cell growth, promote electrical signal conduction, and preserve electrical coupling between the myocardium cells [45,46,47]. The incorporation of GOBMs could treat various cardiac disorders, including cardiac regeneration after myocardial infarction (MI) [31,48], conduction disorders, and restoration [31,46,48] Particularly in the case of MI, GOBMs can overcome the high oxidative stress in the infarcted tissue due to antioxidant activity (Figure 3B) [27]. A combination of other biomaterials, mostly GO and rGO, could be used for developing blood vessels [49] and heart valves [50] to overcome the malfunctions and seems to be an attractive alternative to mechanical and biological prostheses due to high durability, high biocompatibility, and being hemodynamic. Through electrospinning, random and aligned nanofibrous matrices can be fabricated to enable the investigation of isotropic conductivity [48,51]. To this aim, blends of silk fibroin with GO (SF/GO) and rGO (SF/rGO) were prepared for random and aligned electrospinning in a previous study, which resulted in superior electrical conduction properties in SF/rGO (from a mean resistance of 4866.7 MΩ to 4.3 MΩ) in a concentration-dependent manner. The aligned nanofibers not only caused anisotropic conductivity but also aligned and strengthened sarcomeric structures, with an optimum rGO thickness of ~100 nm. The concentration of 0.02–0.08 mg/mL rGO fundamentally enhanced the expression of cardiac-specific proteins, the formation of gap junctions, beating rate, cardiac tissue contraction, and regeneration [48]. Micro- and nanopatterning is another method to develop anisotropic conductive scaffolds to enhance the biomimicry of cardiac tissue constructs. Smith et al. transferred GO films onto polymeric, polyethylene glycol (PEG), and topographic substrates [52]. The electroconductive scaffold had a 7.071 ± 0.124 KΩ resistance in the transverse orientation, which indicated an increment in the myofibril and sarcomere property, cell–cell coupling and calcium-handling protein expression, and action potential [52]. In several studies, other conductive scaffolds (i.e., hydrogels, microgels, sponges, and foams), with a combination of GOBMs, fabricated and fundamentally improved cardiac regeneration and function [27,31,46,53].

### 4.2. Bone Tissue Engineering

The bone has a prominent ability to regenerate; however, the human skeleton has a limited capacity of self-regeneration when bone defects are large enough or critical sized. Several different approaches have been utilized for bone tissue engineering (BTE); the most common were those that tried to mimic the natural process of bone repair using 3D osteoconductive scaffolds. Recently, the potential of GOBMs has gained tremendous attention to facilitate and improve BTE [54,55,56,57,58,59] in its various forms, namely scaffolds, coatings, guided bone membranes, and drug delivery systems [60]. Moreover, GOBMs, particularly with low oxygen contents, have a pivotal influence on adult mesenchymal stem cells (MSCs) to increase osteoinductivity and osteoconductivity [11,61]. Bone is a piezoelectric tissue; hence, using GOBMs, they supply either essential mechanical and biological requirements or electrical conductivity [28]. GOBMs, rGO in particular, could enhance levels of biocompatibility, alkaline phosphatase activity, and calcium deposits, which are essential for bone regeneration [62]. Biphasic calcium phosphate (BCP) coated with rGO (BCP-rGO) at various concentrations was fabricated in a previous study. Briefly, the prepared bone graft materials were implanted in rats with calvarial defects, the common model for investigating new bone regeneration (Figure 4B). The grafting resulted in a new volume of bone formation after eight weeks, particularly in a specimen with a concentration ratio of 4:1000 rGO:BCP (7.65 ± 1.39 mm^3^). The results indicated rGO enhances bone regeneration, although when the rGO percentage exceeded a certain threshold level (≥100 μg/mL), it became severely cytotoxic, [63] which complies with other studies [11,28,60,64]. Recently, researchers have developed 3D-printed scaffolds entailing GOBMs to mimic not only bone geometry but also bone remodeling. Wang et al. fabricated a novel 3D-printed scaffold made of poly(ɛ-caprolactone)/graphene. As expected, the addition of small concentrations of GO enhanced mechanical properties and osteogenesis (by applying electrical stimulation, 10 μA), which led to a better bone defect treatment and bone deposition [28]. Additionally, GOBMs (GO and rGO in particular) presented antimicrobial activity without compromising osteoblasts’ viability, attachment, and proliferation (Figure 4A) [58,65]. A key characteristic of GOBMs is the ability to induce bone differentiation, which leads to rapid bone repair. It has been reported that bioinspired composite scaffolds based on gelatin and GO (GG) could induce the bidirectional differentiation of bone marrow stromal cells (BMSCs) by activating the Erk1/2 and AKT pathway. The presence of GG created the initial hypoxia, which gradually transformed into a well-vasculature robust condition with the bidifferentiation of BMSCs in calvarial defect models [66].

### 4.3. Skin Tissue Engineering

The skin acts as a vital barrier, blocking the entrance of dangerous foreign matter into the body [67]. Because of this significance, GOBMs have been applied for skin regeneration by wound dressings [68] or skin monitoring via electronic skin [69]. Wound dressings, in the form of electrospun mats [70,71], hydrogels [68,72], and sponges, [73] are the best option to accelerate wound healing; hence, GOBMs have been incorporated within these forms of dressings, although other shapes (e.g., modified rGO nanosheets [74]) could also provide drastic novel treatment options. Tang and coworkers fabricated an inspired scaffold by mussel chemistry, based on polydopamine-rGO (pGO) that was incorporated in chitosan (CS) and silk fibroin (SF) hydrogel (pGO-CS/SF). Due to the pGO, electroactivity responded to electrical signals and increased cytological behavior (Figure 5A) [68]. Antioxidant activity reduced cellular oxidation by removing excessive radical oxygen species (ROS) [68,75]. It is a plus that rGO could substantially enhance angiogenesis, which is massively beneficial for slow-healing/chronic wounds [76]. Alongside these applications, it is necessary to recognize the GOBMs and their interactions with the skin, as it is one of the main exposing routes of materials [67,77]. The effects of GO and few-layer graphene (FLG) with HaCaT keratinocytes were investigated. FLG (>5 µg/mL) and mainly GO (~2–5 µg/mL) in long exposure times increased the level of ROS and led to mitochondrial and plasma membrane damages [67,77]. Our own experience with skin regeneration is the development of a membrane from an FGO-based nanocomposite with a hydrophobic outer layer with low porosities to keep bacteria away from the skin, while the inner layer was hydrophilic and more porous to seed with stem cells and growth factors.

### 4.4. Cartilage Tissue Engineering

Cartilage is an avascular tissue that has limited self-healing potential compared to other tissues [78]. Thus, utilizing a proper microenvironment for chondrogenic differentiation will eventually lead to a hyaline-cartilage-like formation. GOBMs in cartilage tissue engineering (CTE) are recognized as promising substitutes due to their self-lubricating and antiwear properties, as well as enhanced mechanical properties, in addition to improved biological behavior [79]. Chitosan/poly(vinyl alcohol) (PVA)/GO (6 wt %) electrospun mats with a 1.81 MPa tensile strength are considered promising when used to reinforce scaffolds in CTE [80]. Yeqiao et al. fabricated a PVA/GO-PEG nanocomposite hydrogel using the freezing/thawing method. The presence of PEG caused the efficient grafting of PVA molecules on the GO surface. The tensile strength, elongation at break, and compressive modulus increased. Furthermore, for samples with a 1.5% GO content, the maximum force retention and dynamic stiffness were improved. In composite hydrogel, the friction coefficient decreased by more than 50% [79]. The small amount of nanographene oxide (NGO) (1 wt %) enhances both the mechanical and biomedical features of microplasma crosslinked gelatin-based hydrogel and provides hyaline cartilage formation without expensive growth factors (Figure 5B). Gelatin methacrylate (GelMa)-poly(ethylene glycol) diacrylate (PEGDA)-GO is a novel cartilage printing ink that increased glycosaminoglycan and collagen levels after the GO-induced chondrogenic differentiation of hMSCs [81]. Seifalian and coworkers have been working on the development of the ear for auricular cartilage reconstruction for children with microtia [82]. Recently, they used an FGO-based nanocomposite as a 3D microporous scaffold with adipose-derived stem cells [83].

### 4.5. Dental Application

Most studies in dental tissue engineering have emphasized the role of GOBMs in the osteogenic differentiation of stem cells after dental implantation, which is an essential function in regenerative dentistry. The potential of GO was demonstrated to induce the odontoblastic or osteogenic differentiation of dental pulp stem cells (DPSCs) without the use of any chemical inductors [84]. However, GOBMs are also utilized in the inhibition of bacterial biofilm formation, membranes, resin, cement, and teeth whitening [85]. Ti, as a metal-based implant, is the best replacement for teeth due to its excellent biocompatibility, high corrosion resistance, and long-term performance; however, Ti has a relatively weak shear strength. The surface treatment of Ti with GOBMs enhances the mechanical implant properties and also exhibits a uniform and widely spread roughness, which affects stem cell [81,85,86,87] bacterial contamination, causes impaired osteogenesis, and is an essential challenge after dental implantation [88]. The presence of GO, coated on Ti(GO-Ti), has shown enhanced antibacterial properties. Immobilized GO on Ti limits the inhibition growth of bacteria, although the addition of antibiotics (e.g., minocycline hydrochloride) or silver nanoparticles is recommended [79,86] (Figure 6). Compared with Ti-based implants, the combination of zirconia (ZrO_2_) and GO in dental implants resulted in healthier surrounding tissue and fewer bacteria growth, in addition to possessing enhanced mechanical features (bending strength increased by 200%, toughness increased by 41%) [89]. The combination of GO with resins, cement, and adhesives inhibits bacterial growth in the oral cavity [85]. Barrier membranes are used in oral surgical procedures for bone augmentation. Membranes can be made of different materials, such as collagen. The presence of GO on the collagen membrane affects the mechanical properties that promote osteoblastic differentiation and decrease inflammation [87]. The last application of GOBMs in the dental field is in teeth whitening, which is based on hydrogen peroxide (H_2_O_2_) and GOBM nanocomposites. This combination enhanced the whitening effect of H_2_O_2_ and decreased treatment time [85].

## 5. Conclusions

Graphene oxide has raised considerable interest in tissue engineering and regenerative medicine due to its specific characteristics, e.g., exceptional mechanical properties and electrical conductivity, as well as physiochemical, antibacterial, and biological capabilities. The biological features of GOBMs depend on many parameters, including concentration, size, shape, and surface chemistry. Within GOBMs, FGO and rGO exhibited the greatest antibacterial capability. Moreover, GOBMs in various forms of 2D and 3D structures could stimulate the proliferation and differentiation of cells to a specific lineage, designated to chemical reactions with biomolecules. Growth factors, small molecules, and ECM proteins force cell differentiation via GOBMs’ π–π and hydrophobic interactions. The main challenges associated with utilizing GOBMs are toxicity, the lack of a detailed mechanism of GOBMs’ biological features, and the explanation of the biological pathway, which has still not been clarified in depth. Moreover, most research has concentrated on toxicity at the cellular level instead of at the genetic level. GOBMs can interact with different biomolecules, especially the DNA; therefore, studies are needed to elucidate the apoptosis pathway. ROS production, which causes cell death, has been studied more, along with lung, liver, intestine, and kidney tissues, than other tissues, e.g., nerve, muscle, cardiac, bone, skin, cartilage, and dental tissue. The application of GOBMs in biomedicine is summarized in Figure 7.

## 6. Future Direction

The long-term toxicity of GOBMs, known as a primary hurdle, has limited experimental data, and further research investigating their effects is necessary (Figure 8). This uncertainty has impeded widespread access to clinics; therefore, ensuring human safety is a priority for these biomaterials. In other words, to achieve the best clinical outcomes, it is essential to consider solutions to overcome the possible challenges of GOBMs, such as their toxicity and biodegradability; for example, by developing standardized parameters for GOBMs utilization, as their toxicity and biological features are intertwined with physicochemical parameters. Another solution that could minimize these drawbacks is fabricating biocomposites. It is worth noting that the dose- and time-dependent effects of GOBMs dictate whether or not ROS scavenging or oxidative stress affects cells detrimentally. The interaction of GOBMs at the genetic level should be considered as well, as they can interact with DNA and result in alterations of the genetics in the human population. Although research on GOBMs is still in an early stage, due to its versatile properties, it will provide extensive applications for biomaterial science and regenerative medicines. We have been working on the development of functionalized graphene oxide (FGO)-based nanocomposite materials for biomedical applications [82]. The two families of materials developed are Hastalex™, a nonbiodegrable nanocomposite for applications such as heart valves [50], tendons, and breast implants, and a bioabsorbance version (BioHastalex™), which has been used for nerve regeneration and tissue engineering applications, as well as for other industries such as the textile industry, with sustainably produced, nontoxic, and biodegradable materials in the ocean (Nanoloom.co.uk).

## Figures and Tables

**Figure 1 nanomaterials-11-01083-f001:**
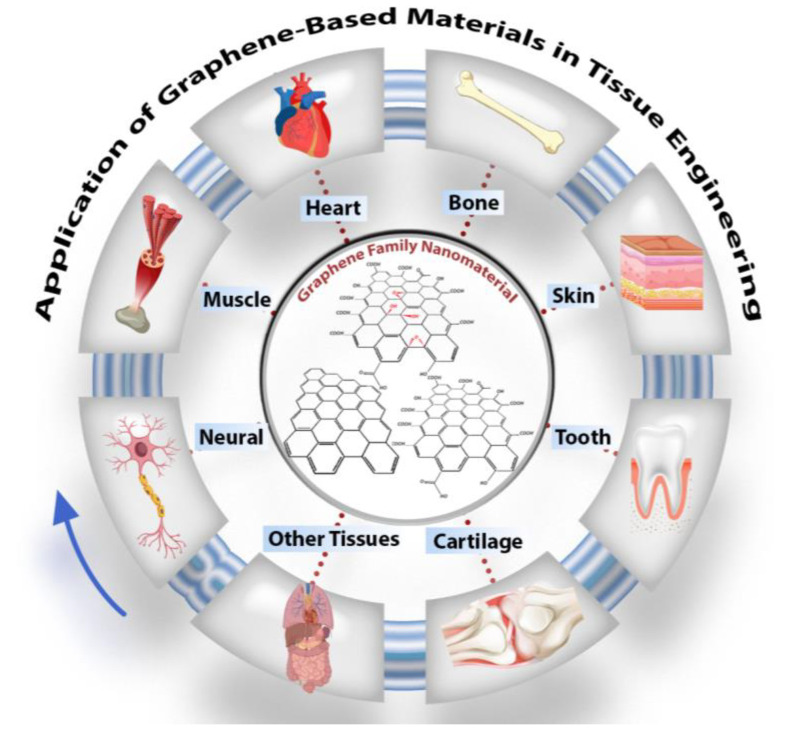
Schematic of graphene oxide nanomaterials and their application in tissue engineering, particularly in nerve, muscle, heart, skin, cartilage, dental, and other tissues.

**Figure 2 nanomaterials-11-01083-f002:**
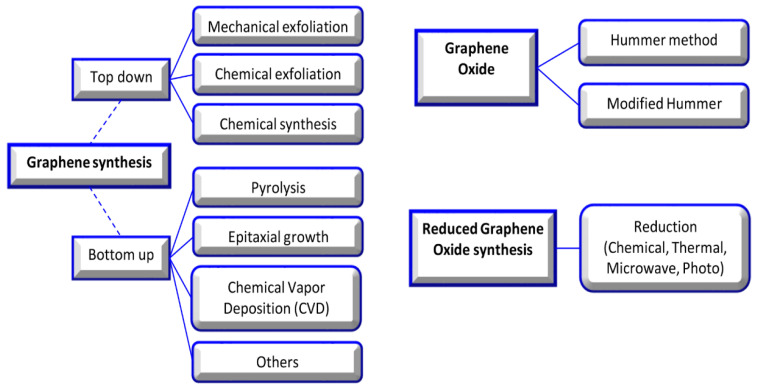
Graphene, graphene oxide, and reduced graphene oxide synthesis methods [9].

**Figure 3 nanomaterials-11-01083-f003:**
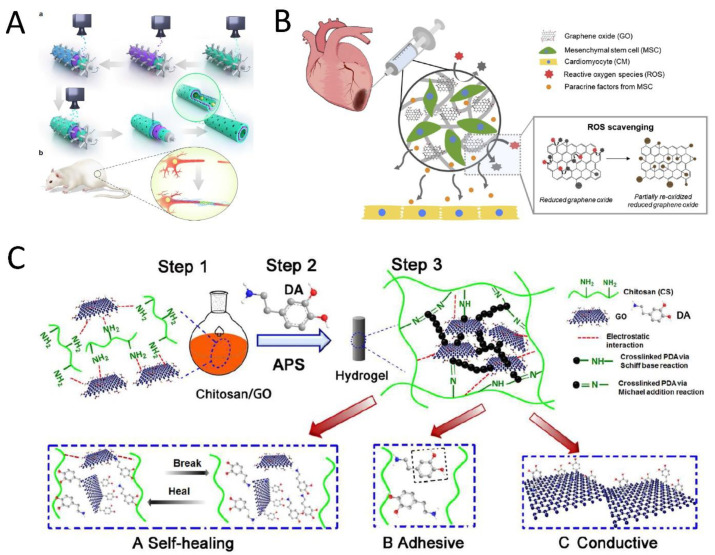
Schematic illustration of (**A**) the LBLC graphene nerve conduit. (a) The green layers are PDA/RGD adhesive macromolecules. The purple layer is single or multilayered graphene and the PCL-blended layer. The blue layer is the graphene and PCL-blended layer once more. (b) The GO/PCL nerve conduit in a sciatic nerve defect model in SD rats. Reused with permission from [30]. Copyright © 2021, Springer Nature. (**B**) rGO/alginate microgel embedding MSCs for cardiac tissue repair post-MI. Due to the loaded rGO, it was expected that encapsulated MSCs were protected from the severe oxidative stress in the infarcted tissue and facilitated cardiac regeneration. Reused with permission from [27]. Copyright © 2021, Elsevier Ltd. (**C**) CS-DA-GO composite hydrogel fabrication in three steps, which causes the enhanced properties of the hydrogel. (A) Self-healing mechanism of the hydrogel. (B) Self-adhesiveness of the hydrogel. (C) Enhanced conductivity. Reused with permission from [29]. Copyright © 2021, Elsevier Ltd.

**Figure 4 nanomaterials-11-01083-f004:**
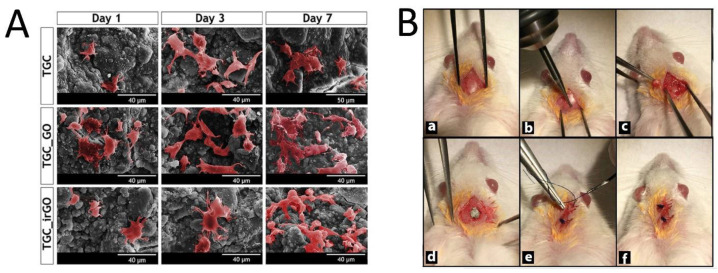
(**A**) Pseudocolored SEM images of human osteoblasts that successfully adhered to the surface of scaffolds at an upward trend (Days 1, 3, and 7). Reused with permission from [58]. Copyright © 2021, Elsevier Ltd. (**B**) Surgical procedure of the preparation of the surgical size defect (5 mm) critical in the rat calvarium (**a**–**c**), placement of the chitosan–graphene oxide scaffold (**d**), and closure of the periosteum and skin (**e**,**f**). Reused with permission from [64]. Copyright © 2021, Springer Nature.

**Figure 5 nanomaterials-11-01083-f005:**
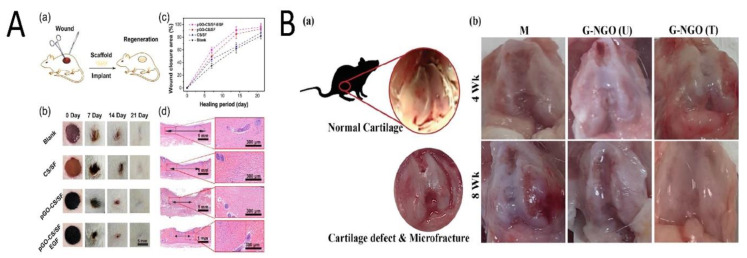
(**A**) Wound healing of the skin through the treatment of various scaffolds. (**a**) Model of wound regeneration. Digital images (**b**) and closure rate (**c**) of the wound defects; multiple treatments at Days 0, 7, 14, and 21. (**d**) Depictive images of H&E-stained histological sections after 21 days (arrows indicate the granulation tissue). Reused with permission from [68]. Copyright © 2021 American Chemical Society. (**B**) (**a**) A normal rat knee joint; (**b**) knee joint restoration in different experimental groups (M: without implant; GO-NGO (U): non-crosslinked hydrogel; GO-NGO (T): microplasma crosslinked hydrogel) i 4 and 8 weeks. Reused with permission from [24]. Copyright © 2021 American Chemical Society.

**Figure 6 nanomaterials-11-01083-f006:**
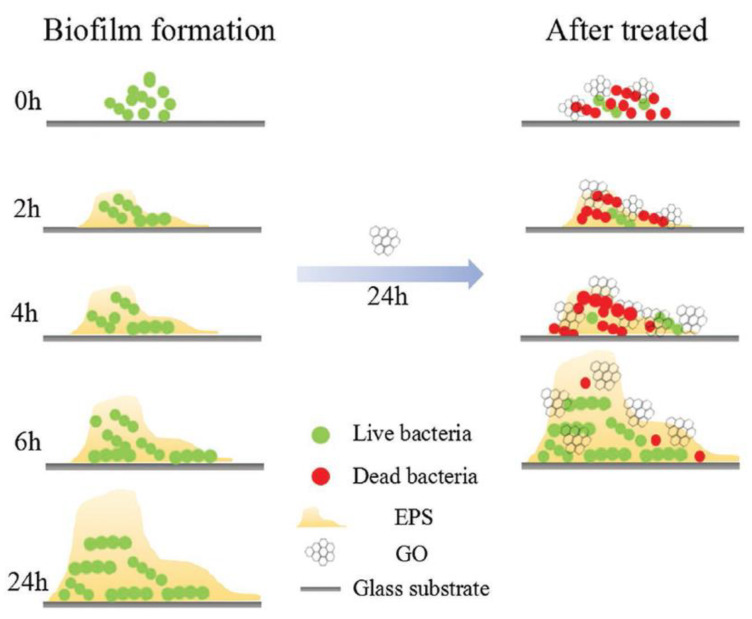
The prohibition effect of GO on biofilm formation. Reused with permission from [90]. Copyright © 2021 WILEY-VCH Verlag GmbH and Co. KGaA, Weinheim.

**Figure 7 nanomaterials-11-01083-f007:**
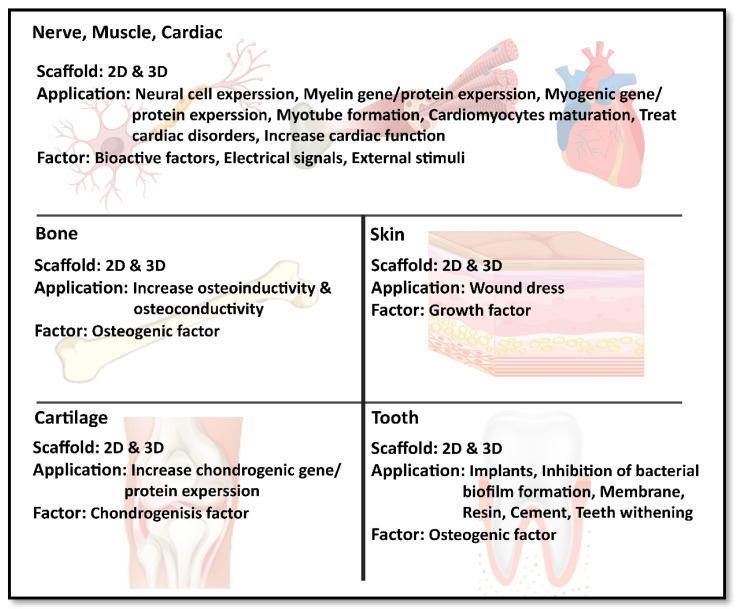
Graphical conclusion.

**Figure 8 nanomaterials-11-01083-f008:**
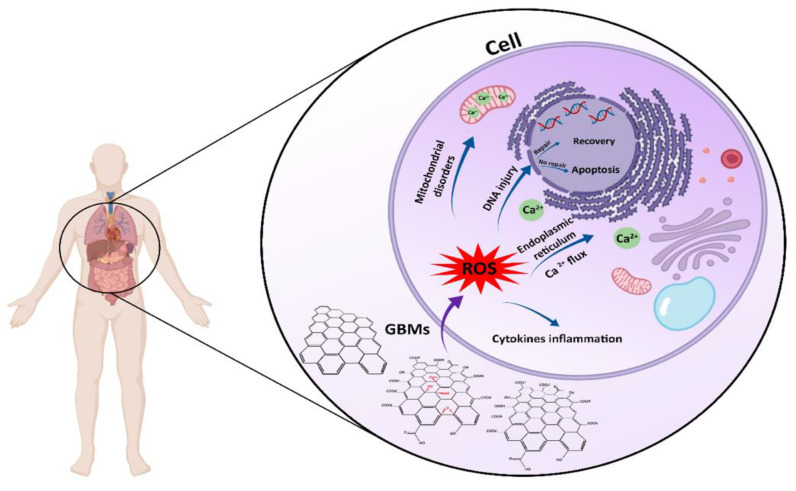
Schematic illustration of the cellular toxicity of GOBMs when exposed to the cell. GOBMs enter the cell through various pathways, which affects their shape, size, and surface chemistry and eventually results in ROS production. An increased ROS level may cause mitochondrial membrane depolarization and Ca^2+^ release; lipid, protein, and DNA damage; and inflammation response by releasing cytokines and chemokines.

**Table 1 nanomaterials-11-01083-t001:** Summary of preclinical studies of graphene-oxide-based materials in tissue engineering from 2017.

Organs	Materials	Animal	Cells/Stem Cells	Experiment	Outcome	Year [Ref] Country
Cartilage	Gelatin (G) and GO	Rats	Human chondrosarcoma cell, rat BMSCs, and rat chondrocyte cells	Preparing nano-GO (NGO) solution => hydrogel crosslinking => three groups: non-crosslinked hydrogel (NGO (U)), crosslinked hydrogel (NGO(T)), control group (G)	NGO(T) vs. NGO(U): ↑ mechanical properties. No significant cytotoxicity After implantation (8 weeks): fibrous tissue repair for U and Complete repair for T	2020 [24] Taiwan
Nervous system	GO, antheraea pernyi silk fibroin (ApF) and PLCL	Rats	Schwann and PC12 cells	Coating GO on ApF/PLCL nanofibers => GO reduction => applying ES => preparing the AP/RGO nerve guidance conduit	↑ CPAM and ↑ MPs. GO: ↑ focal adhesion kinase expression of PC12 cells. ↑ Repair in animal model’s sciatic nerve	2019 [25] China
Muscle	GO, rGO, polyacry- lamide (PAAm)	Mice	C2C12 myoblasts	Incorporating GO into PAAm (GO-PAAm) => micropatterning of GO-PAAm with femtosecond laser ablation (FLA) => production of micropatterned conductive r(GO/PAAm)	Micropatterned: ↑ differentiation and myoblast alignment r(GO/PAAm) vs. GO/PAAm: ↓ impedance values PD50/r(GO/PAAm) (optimum): ↑ tissue compatibility and ES => ↑ myogenesis.	2019 [26] Korea
Heart	GO, rGO and alginate	Rats with MI	Human mesenchymal stem cells	GO/Ag blend => hMSCs encapsulation => electrospraying and then crosslinking => GO/Ag microgels => reductive treatment => r(GO/Ag)	rGO vs. GO: ↑ CPAM, ↑ antioxidant activity, and ↓ Oxidative stress hMSCs-CMs vs. CMs: ↑ cell viability and cardiac maturation => expressing cardiac markers	2019 [27] Korea
Bone	GO and poly(ɛ-caprolactone)	Rat	MC3T3 preosteoblastic cells	Synthesizing GO => PCL/GO pellets => melt blending => 3D printing	↑ Protein absorbent and ↑ CPAM ↓ Immunogenicity. Treating a rat calvaria critical size defect => well-organized tissue deposition and bone remodeling.	2019 [28] UK
Skin	Polydopamine (P), rGO (pGO), chitosan (CS), and silk fibroin (SF) (pGO-CS/SF)	Rats with a full-thickness skin defect	RAW 2467 cells and C2C12 myoblast cells	Dispersing pGO into CS/SF mixture=> dual-crosslinking by poly(ethylene glycol) diglycidyl ether (PEGDE) and glutaraldehyde (GA) => freeze-drying => pGO-CS/SF scaffold	↑ CPAM and ↑ mechanical properties ↑ Antioxidant activity => reduce cellular oxidation. Well-connected electric pathway ↑ Wound healing and ↓ oxidative stress and inflammatory responses	2019 [29] China
Nervous system	Single (SG) and multilayered (GM) graphene PCL, RGD, polydopamine	Rats	Schwann cells	Fabricating nanoscaffolds => seeding Schwann cells => implanting the 3D scaffold in sciatic nerve defect models.	↑ CPAM and ↑ neural cell expressionPDA/RGD-SG/PCL and PDA/RGD-MG/PCL nerve conduits: ↑ neural regeneration	2018 [30] China
Heart	GO, gold nanoparticles (AuNPs) (GO-AuNPs), chitosan (CS)	Rats with MI	Rat smooth muscle cells, mouse fibroblasts, and human iPSC-CMs	GO => embedding with AuNPs by thermal-reduction => GO-AuNPs => CS solution addition and freeze-drying => CS-GO-Au scaffolds	↑ Electrical conductivity (at 0.5% w/v GO- AuNPs). ↑ CPAM, no immune response ↑ QRS interval (by ↑ conduction velocity and ↑ contractility). ↑ Connexin43 (Cx43) ↑ Electrical conduction and ventricular function	2018 [31] Canada
Dental	GO, chitosan (CS), hydroxyapatite (HA) and Titanium (Ti)	Rats	Bone marrow stromal cells (BMSCs)	Coating GO/CS/HA on Ti substrates by electrophoretic deposition (EPD)	↑ CPAM and ↑ osseointegration in vivo	2018 [32] China
Dental	GO and Collagen	Dog	Mouse osteoblastic MC3T3-E1 cells	Coating Ti on the 3D collagen scaffold => evaluation of bone augmentation on the rat cranial bone => assessing the periodontal healing of class II furcation defects	↓ Cytotoxicity. GO: cellular ingrowth behavior and angiogenesis => ↑ rat bone augmentation ↑ Periodontal attachment	2018 [33] Japan
Nervous system	GO, rGO, and Gelatin	Rats	Embryonic neural progenitor cells	Synthesizing rGO microfibers from GO => assembling rGO microfibers into the 3D gelatin hydrogel for stable implantation	Microfiber coated with adhesive molecules => interconnected culture ↑ Differentiation in the defect site	2017 [34] Spain
Bone	GO and chitosan (CHT)	Mice	Murine preosteoblasts belonging to the 3T3-E1 cell line	CHT/GO blend => freeze-drying	↑ Alkaline phosphatase activity (ALP), ↑ osteogenesis, and ↑ bone morphogenetic protein expression ↑ Differentiation of osteoprogenitor cells ↑ New bone formation	2017 [35] Romania
Bone	rGO and nanohydroxyapatite (nHA)	Rabbits	Bone mesenchymal stem cells	Self-assembling of GO and nHA => nHA@RGO	↑ CPAM, ↑ ALP, and ↑ osteogenic gene expression ↑ Healing circular calvarial defects (optimum: 20% nHA@RGO) ↑ Collagen deposition and ↑ mineralization	2017 [11] China

Keys: **↑**, increased or improved; ↓, decreased; =>, followed by; CPAM, cellular proliferation, adhesion, and migration; SD, Sprague–Dawley; MPs, mechanical properties; HM, Hummer’s method; ES, electrical stimulation.

**Table 2 nanomaterials-11-01083-t002:** The most common stem cells utilized in each tissue [24,26,32,90].

Tissue	Stem cells
Bone	hMSCs, hADMSCs, MC3T3-E1, DPSCs, PDLSCs
Nerve	NSCs, hMSCs, hADMSCs, ESCs, iPSCs, SCAP
Muscle and cardiac	C2C12, MSCs, hMSCs, cardiomyocytes, and EC
Cartilage	Human mesenchymal stem cell
Skin	MSCs, human dermal fibroblasts (HDFs)
Dental	DPSCs, PDLSCs, hMSCs, BMSCs

Keys: hADMSCs, human adipose-derived mesenchymal stem cells; DPSCs, dental pulp stem cells; PDLSCs, periodontal ligament stem cells; NSCs, neural stem cells; ESCs, embryonic stem cells; iPSCs, induced pluripotent stem cells; SCAP, stem cell from apical papilla; BMSCs, bone-marrow-derived stem cells; C2C12, mouse myoblasts; EC, endothelial cells.

## Data Availability

Some data available on www.NanoRegMed.com (accessed on 21 April 2021).

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
