# Peer review of "Graphene Oxide: Opportunities and Challenges in Biomedicine"

_nanomaterials, 2021, doi:10.3390/nano11051083_

Round 1

Reviewer 1 Report

Dear authors, I analyzed carefully your work and I could say that this could be a good article, but unfortunately you have performed 3 main mistakes:

  • In fact, graphene is present in the title only, all described properties concern GO or its derivatives; so please consider the change of it, perhaps “Graphene Derivatives: Opportunities and Challenges in Biomedicine” would be more suitable. More, GO is not in fact the graphene-, but rather graphite-derivative.
  • There is a lot of strange set of sentences: e.g. p2, l.61-63, p2 l 76-80, p3. L.97 (“even”? maybe “especially”), p3.l.112-115; etc. etc.
  • it would be good to add some feelings into the text, there is a lot of opportunities to break a routine review article. In present form it looks like (good) MSC dissertation.

Nevertheless, I would like to commend you for the Figures and Tab.1, good work, congratulations.

Reviewer 2 Report

The paper «Graphene and its Derivatives: Opportunities and Challenges in Biomedicine» is an interesting review focused on application of graphene and its modification in development of tissues and organs. The list of references contains 84 papers and all of them were published in last 4 years. The article is well-organized. It’s good that authors state common principles of graphene application for each of considered tissues and organs before starting the review. I’d like to thank authors for their job.

To my mind, analysis of some additional papers would improve the review:

  • Graphene in nerves engineering

https://doi.org/10.1016/j.msec.2020.111632

https://doi.org/10.1016/j.jmbbm.2019.103387

https://doi.org/10.1016/j.molstruc.2019.03.058

  • Graphene in bones engineering

https://doi.org/10.1016/j.bioactmat.2020.12.003

https://doi.org/10.1016/j.nano.2020.102251

  • Graphene for dental application

https://doi.org/10.1016/j.ceramint.2020.11.041

https://doi.org/10.1016/j.dental.2016.09.030

Round 2

Reviewer 1 Report

Dear Authors, thank you